# Analysis of Vancomycin-Resistant Enterococci in Hemato-Oncological Patients

**DOI:** 10.3390/antibiotics9110785

**Published:** 2020-11-07

**Authors:** Kristýna Hricová, Taťána Štosová, Pavla Kučová, Kateřina Fišerová, Jan Bardoň, Milan Kolář

**Affiliations:** Department of Microbiology, University Hospital Olomouc and Faculty of Medicine and Dentistry, Palacký University Olomouc, 77900 Olomouc, Czech Republic; hricova.k@email.cz (K.H.); tatana.stosova@fnol.cz (T.Š.); pavla.kucova@fnol.cz (P.K.); jbardon@svuol.cz (J.B.); milan.kolar@fnol.cz (M.K.)

**Keywords:** VRE, GIT, hemato-oncological patients, clonality

## Abstract

Enterococci are important bacterial pathogens, and their significance is even greater in the case of vancomycin-resistant enterococci (VRE). The study analyzed the presence of VRE in the gastrointestinal tract (GIT) of hemato-oncological patients. Active screening using selective agars yielded VRE for phenotypic and genotypic analyses. Isolated strains were identified with MALDI-TOF MS, (Matrix-Assisted Laser Desorption/Ionization Time-of-Flight Mass Spectrometry) their susceptibility to antibiotics was tested, and resistance genes (*vanA*, *vanB*, *vanC-1*, *vanC2-C3*) and genes encoding virulence factors (*asa1*, *gelE*, *cylA*, *esp*, *hyl*) were detected. Pulsed-field gel electrophoresis was used to assess the relationship of the isolated strains. Over a period of three years, 103 VanA-type VRE were identified in 1405 hemato-oncological patients. The most frequently detected virulence factor was extracellular surface protein (84%), followed by hyaluronidase (40%). Unique restriction profiles were observed in 33% of strains; clonality was detected in 67% of isolates. The study found that 7% of hemato-oncological patients carried VRE in their GIT. In all cases, the species identified was *Enterococcus faecium*. No clone persisted for the entire 3-year study period. However, genetically different clusters were observed for shorter periods of time, no longer than eight months, with identical VRE spreading among patients.

## 1. Introduction

In today’s medicine, important bacterial pathogens are enterococci. As etiological agents, they are responsible for community as well as healthcare-associated infections (HAIs), particularly in patients with prolonged hospital stays, severe underlying disease, or previous broad-spectrum antibiotic therapy [1]. Sievert et al. reported that enterococci belong to the most common bacterial pathogens causing HAIs in the USA [2]. According to a 2017 U.S. study, enterococci were implicated in 14% of HAIs in the country [3]. Using results from a large multicenter study involving 15 centers in the Czech Republic, Kolář et al. documented that 9% of bloodstream infections were due to enterococci [4]. Herkel et al. found that enterococci accounted for 5% of all bacterial pathogens causing healthcare-associated pneumonia [5]. In these cases, they originated in the upper gastrointestinal tract (GIT), with pneumonia developing as a result of gastric content regurgitation and subsequent microaspiration [6]. Since approximately the 1970s, a vast majority of enterococcal HAIs have been caused by *Enterococcus faecium* and *Enterococcus faecalis* [7].

Treatment of enterococcal infections is considerably influenced not only by their natural resistance to many antibacterial agents, but also by acquired resistance, representing a factor limiting antibiotic therapy. A therapeutic challenge and real threat to patients are vancomycin-resistant enterococci (VRE). The most widespread VRE type is the VanA phenotype, determined by the D-alanyl-D-lactate structure encoded by the *vanA* gene that is very often located on transposon Tn1546 [8,9,10,11]. The first description of VRE comes from the UK and was published in 1988 [12]. In the Czech Republic, VRE were first detected in hemato-oncological patients at the University Hospital Olomouc in 1997 [13].

At present, a serious public issue is the presence of multidrug-resistant bacteria including VRE in the GIT as components of the normal microflora. The clinical significance of multidrug-resistant bacteria in the GIT was documented by Kolář et al. In their study, an ESBL-positive strain of *Klebsiella pneumoniae* and an AmpC-positive strain of *Enterobacter cloacae*, isolated from rectal swabs and thus being part of gastrointestinal bacterial flora, were found to be identical with strains causing bacterial infections, namely urinary tract infection and bloodstream infection [14]. Arias and Murray reported translocation of enterococci across the GIT wall into the bloodstream with subsequent endocarditis [7]. This fact is of utmost importance in VRE that may be part of GIT microflora, with the length of hospital stay or interventions such as hemodialysis, transplantation, and application of artificial materials playing an important role [15,16,17]. Heisel et al. showed that the proportion of patients with VRE colonization of the GIT changed from 3% before hospital admission to 59% during their hospital stay [18]. Another important factor promoting the presence of VRE in the GIT is the selection pressure of antibiotics. Kolář et al. documented the positive selection pressure of third-generation cephalosporins and glycopeptides with regard to the presence of VRE [19]. Paterson et al. found that after treatment with piperacillin-tazobactam or cefepime, 26% and 31%, respectively, of patients primarily tested negative for VRE were colonized with these bacteria [20].

Data from the EARS-Net provide information on the current prevalence of VRE in the Czech Republic [21]. The percentage of vancomycin-resistant strains of *E. faecium* ranges from 4% to 13% [22,23,24]. However, this prominent European database only considers blood isolates and cannot provide more accurate comprehensive data on the prevalence of VRE in both the community and hospital human populations including GIT carriage. In the Czech Republic, such data were last published by Kolář et al. in 2006 [19,25]. In their study on the prevalence of VRE in hemato-oncological patients in a Czech university hospital, these strains accounted for 5% of all isolated enterococci. The most frequent strains were VanA-type *E. faecium* (78%) and VanB-type *E. faecalis* (10%). The fact that VRE were most frequently isolated from rectal swabs (55%) illustrates the significance of their carriage in the GIT [19].

A key role in the etiopathogenesis of infections caused by enterococci may be played by selected virulence factors. These are essential for biofilm formation and subsequent colonization [26]. The presence of selected virulence factors in VRE was studied by Yang et al. For *E. faecium*, the most frequent gene was *esp*, encoding extracellular surface protein, detected in 90% of strains. Additionally present (28%) was the *hyl* (hyaluronidase) gene. The combination of these two genes (*esp* + *hyl*) was noted in 26% of isolates. Genes encoding aggregation substance (*asa1*), cytolysin (*cylA*), or gelatinase (*gelE*) were not detected in *E. faecium*, but only in *E. faecalis* strains, which were considerably less frequently detected in the study [27].

The present study aimed to determine the presence of VRE in the GIT of hemato-oncological patients in the University Hospital Olomouc and to carry out their genetic analysis including the assessment of selected virulence factors.

## 2. Results

Over a 3-year study period, stool samples and perianal swabs obtained from 1405 subjects were analyzed and a total of 103 VRE were isolated, suggesting that 7.3% of hemato-oncological patients carried VRE in their GIT. In all cases, the species identified was *E. faecium*.

The results of VRE susceptibility testing showed very high susceptibility to linezolid (100%) and tigecycline (96%) and, conversely, 100% resistance to ampicillin and teicoplanin (Table 1).

In all isolated VRE, the *vanA* gene was present. Strains with VanB, VanC-1, and VanC2-C3 phenotypes were not detected.

Analysis of the selected virulence factors showed that extracellular surface protein (*esp*) was most frequent. It was present in 84% of isolates, either alone (53%) or in combination with hyaluronidase (*esp* + *hyl*) in 31% of isolates. Hyaluronidase (*hyl*) was found in 40% of isolates (alone in 9%). No virulence factor was present in 7% of isolates.

Based on the pulsed-field gel electrophoresis (PFGE) results, a dendrogram was produced, as shown in Figure 1. Among the 103 VRE, a total of 69 strains were identified and distributed into 18 groups (A–R) based on their similarity (tolerance 1.5; cut-off, 95%). Thus, the overall clonality was 67%. The remaining 34 strains (33%) had unique restriction profiles. Within individual clonal groups, isolates showed 98% similarity. Clone G was the largest cluster, containing 12 strains. Genotypes A, E, F, H, I, J, O, P, and Q were relatively small, all containing two strains. Clones D, N, and R contained three strains each. Five strains in Clone M were isolated from patients staying in the same ward over a period of one month. Analysis of the time–spatial relationships of these patients confirmed transmission of a single VRE clone, but failed to identify the index patient or other source of the outbreak. Clone G consisted of 12 strains isolated from patients over a period of eight months. Clone K consisted of eight strains, all of which were isolated over a period of three months, namely September–November 2017. Similarly, Clone C consisted of eight strains, isolated over four months (September–December 2018). Clone B contained five strains isolated over three months (January–March 2017) and clone L consisted of fours strains obtained over six months in 2018. The timeline of the prevalence of VRE clones throughout the study duration is shown in Figure 2.

## 3. Discussion

Infections caused by VRE are often preceded by gastrointestinal colonization with these significant bacterial pathogens. A meta-analysis of 45 studies comprising over 8000 patients with malignancies showed approximately 20% colonization of the GIT with VRE [28]. In a 2019 study on GIT colonization with resistant bacteria in patients with acute myeloid leukemia, Ballo et al. even detected a much higher rate of 74% VRE colonization [29]. In the same year, another study found that VRE-colonized patients were as much as 24 times more likely to develop bloodstream infections caused by these strains [30]. Of importance is also the increase in the prevalence of VRE in the GIT of nursing facility patients from 10% to 29%, with MRSA detection rates declining from 37% to 13% over the 2003–2016 period [31]. In the present study, VRE were detected in the GIT of only 7% of hemato-oncological patients. This may be explained by very low prevalence of VRE (below 1%, unpublished data) in the GIT of people living in the community in the region from which the monitored patients come. Another important reason is the implementation of bacteriological surveillance in hemato-oncological patients including active search for bacterial strains with dangerous resistance phenotypes (e.g., VRE, ESBL-, AmpC- and KPC-positive enterobacteria) in the microflora of the upper respiratory tract and GIT. In the case of positive detection, barrier measures were strictly observed in these patients.

A study by Pudová et al. showed that the GIT is an important source of enterococci causing healthcare-associated pneumonia [6]. If VRE were implicated, the situation would be much more serious due to potential failure of initial antibiotic therapy. Therefore, these dangerous bacteria need to be detected in the GIT of patients, especially immunocompromised ones. All VRE isolated in our study belong to the VanA phenotype of *E. faecium* strains. In a 2006 study conducted in the same department, genetic analysis of VRE revealed that in hemato-oncological patients, the most frequent source was the GIT (55%) and the strains were mostly VanA-positive *E. faecium* (78%) and VanB-positive *E. faecalis* (10%) [19]. 

In Europe, the most frequent VRE phenotype is VanA [32]. This was confirmed by the results of genetic analysis of VRE in the present study showing that the *vanA* gene was present in 100% of the tested isolates while the *vanB*, *vanC-1*, and *vanC2/C3* genes were completely absent. This is consistent with results from a 2018 Serbian study by Jovanović et al. investigating VRE isolated from the oropharynx and stools of hemato-oncological patients. They also showed 100% presence of the *vanA* gene and 0% presence of the *vanB* gene [33].

Virulence factors are likely to play an important role in the etiopathogenesis of enterococcal infections [34,35]. These factors contribute to colonization, biofilm formation, and the production of enzymes potentially increasing the severity of developing infections [7]. In the present study, selected virulence factors were identified in 93% of isolated VRE. The presence of the most frequent virulence factor (*esp*) was lower than that in a similar 2018 study [36]. While Marchi et al. reported 97% prevalence of the *esp* gene in VRE, there were 84% of *esp*-positive VRE in hemato-oncological patients in the present study. Molecular analysis of virulence factors in VRE isolated from oncological patients is available in a 2013 Mexican study. Compared to our study, the *esp* gene was less frequent, being present in 50% of VRE isolates. The *hyl* gene was detected in 17% of cases compared with 40% in the present study. The *esp* + *hyl* gene combination was equally present in both studies, namely in 31% in the present study and in 33% in the study by Ochoa et al. [37]. Comparison of the presence of virulence factors between individual VRE clones did not show significant differences.

Even though VRE are widespread globally, their epidemiology varies by region. The spread of VRE between various hospitals was illustrated by a study from Ireland that had long faced high VRE prevalence rates, reaching approximately 45% [24,38]. Recurring VRE outbreaks remained an unresolved problem in the healthcare system. Only adequately selected molecular typing methods may confirm or rule out epidemiologically related cases. If a new outbreak or merely increased rate of VRE is reasonably suspected, the clonal relationship of strains needs to be analyzed to reveal the source or route of transmission of the infection. In the present study, the clonal relationship of strains was assessed with PFGE. Despite certain limitations, PFGE continues to be considered a gold-standard method as its results may be quantified with high discriminatory ability and reproducibility [39,40]. In the present study, molecular typing results confirmed clonal spread of identical strains in 67% of tested VRE isolates. This confirms the possibility of colonization in staff members and subsequent transmission to hospitalized patients via various environmental vectors. Apparently, the hands of the staff members are the most important one. The literature also mentions wastewater in a commode as an appropriate vehicle for VRE survival, enabling spread in the form of aerosol or by contact with hands or clothes [41]. With regard to the finding of unique profiles in 33% of analyzed VRE, their endogenous origin from the GIT is to be admitted, with the following selection caused by broad-spectrum antibiotic therapy.

Analysis of the restriction profiles of individual clones yielded clonal groups containing two to 12 strains obtained from various patients. The study results suggest that no single epidemic clone was spreading at the hemato-oncology department for a longer period of time and that individual epidemic clones were present for limited time intervals of one to eight months. It can be assumed that the time of occurrence of individual VRE clones was mainly conditioned by the length of hospital stay. There were no changes in the environmental parameters during the study (e.g., remodeling of the rooms or new ventilation).

## 4. Materials and Methods

Between 1 January 2016 and 31 December 2018, stool samples and perianal swabs were collected from patients staying at the Department of Hemato-Oncology, University Hospital Olomouc to isolate enterococci using standard procedures. All samples were inoculated onto both blood agar plates and Brilliance VRE Agar chromogenic screening plates (Oxoid, Brno, Czech Republic). Species identification of isolates was carried out with MALDI-TOF MS (Biotyper Microflex, Bruker Daltonics, Billerica, MA, USA). From each patient, only one isolate identified as the first one was included. Strains were stored in cryotubes at −80 °C (Cryobank B, ITEST, Hradec Králové, Czech Republic). Stool samples and perianal swabs were collected only as part of standard clinical care.

Susceptibility to antibiotics (ampicillin, tigecycline, tetracycline, teicoplanin, nitrofurantoin, and linezolid) was tested using the microdilution method as recommended by the EUCAST [42]. *Staphylococcus aureus* ATCC 29213 and *Enterococcus faecalis* ATCC 29212 reference strains were used for quality control.

DNA for genetic analysis was isolated with a Qiagen Kit (DNeasy Blood and Tissue Kit, Qiagen, Hilden, Germany). A PCR (polymerase chain reaction) method was used to detect glycopeptide resistance genes (*vanA*, *vanB*, *vanC-1*, *vanC2-C3*) with primers as specified in Table 2 [43]. The amplification conditions were as follows: initial denaturation at 95 °C for 7 min, followed by 35 denaturation cycles at 95 °C/30 s, annealing at 62 °C/30 s, extension at 72 °C/60 s, and final extension at 72 °C for 7 min. Selected virulence factors were identified with multiplex PCR [44]. The presence of genes encoding gelatinase (*gelE*), aggregation substance (*asa1*), hyaluronidase (*hyl*), cytolysin (*cylA*), and extracellular surface protein (*esp*) was investigated using the primers listed in Table 2. The reaction was run under the following conditions: initial denaturation at 95 °C for 15 min, followed by 30 denaturation cycles at 95 °C/60 s, annealing at 62 °C/60 s, extension at 72 °C/60 s, and final extension at 72 °C for 10 min. The final products were separated in 1.5% agarose gel at 110 V for 60 min using SYBR Safe DNA Gel Stain (Thermo Fisher Scientific, Waltham, MA, USA). Visualization was carried out under UV light with a UV transilluminator (Transluminator Discovery^TM^, UltraLum, Claremont, CA, USA).

Molecular typing of all isolated VRE was performed using PFGE. Intact DNA was obtained according to a previously described protocol [45]. Restriction cleavage of blocks with DNA was carried out in a *SmaI* restriction enzyme solution (Takara Biotechnology, Kyoto, Japan) containing 5 μL of *SmaI* restriction buffer, 5 μL of 0.1% BSA, 50 μL of deionized water, and 10U of *SmaI*. The blocks were incubated at 30 °C overnight. Restriction fragments were separated in 1.2% (*w*/*v*) agarose gel in the CHEF-DRII system (Bio-Rad, Hercules, CA, USA). The parameters selected for analysis were as follows: 24 h, voltage 6 V·cm^−1^, and pulse times 2–35 s. The restriction fragments were assessed with GelCompare, version 2.0 (Applied Maths, Kortrijk, Belgium). The similarity coefficient was calculated using the Dice algorithm (based on the Dice coefficient of similarity of macrorestriction profiles set at 2%). Individual clusters were analyzed with the UPGMA (unweighted pair group method with arithmetic mean) algorithm and the results were interpreted using criteria defined by Tenover et al. [46].

## 5. Conclusions

The results of PFGE showed no significant outbreak of clonal spread of VRE, but confirmed repeated smaller outbreaks. As seen from the constructed timeline, there was no single hospital strain with an unchanged genetic profile to repeatedly occur over the three-year study period. During that time, a total of 18 clonal groups were identified. In 2016, an identical VRE clone detected in 12 patients persisted for as long as eight months. The year 2017 seems to be most diverse, with only 1-month persistence of a clone replaced by a genetically different clone being noted in four cases. The number of clone groups in the last year was comparable to that in 2017 as it only decreased by one group. However, identical strains persisted for a longer time in various patients and no clone persisted for less than three months. Moreover, the study revealed the presence of genetically unique isolates accounting for 33% of all VRE. One of the risk factors for colonization is the long-term hospital stay of patients sharing the same room combined with the ability of enterococci to persist in the hospital environment [19]. It is those small groups of clones identified in two or three patients in the present study that suggest their spread among patients hospitalized at the same time. The potential routes of transmission are the hands of both patients and healthcare professionals or even family members visiting patients. Considering the selection pressure of antibiotics used, the immune status of hemato-oncological patients and chemotherapy, if administered, the likely source of VRE is their GIT.

## Figures and Tables

**Figure 1 antibiotics-09-00785-f001:**
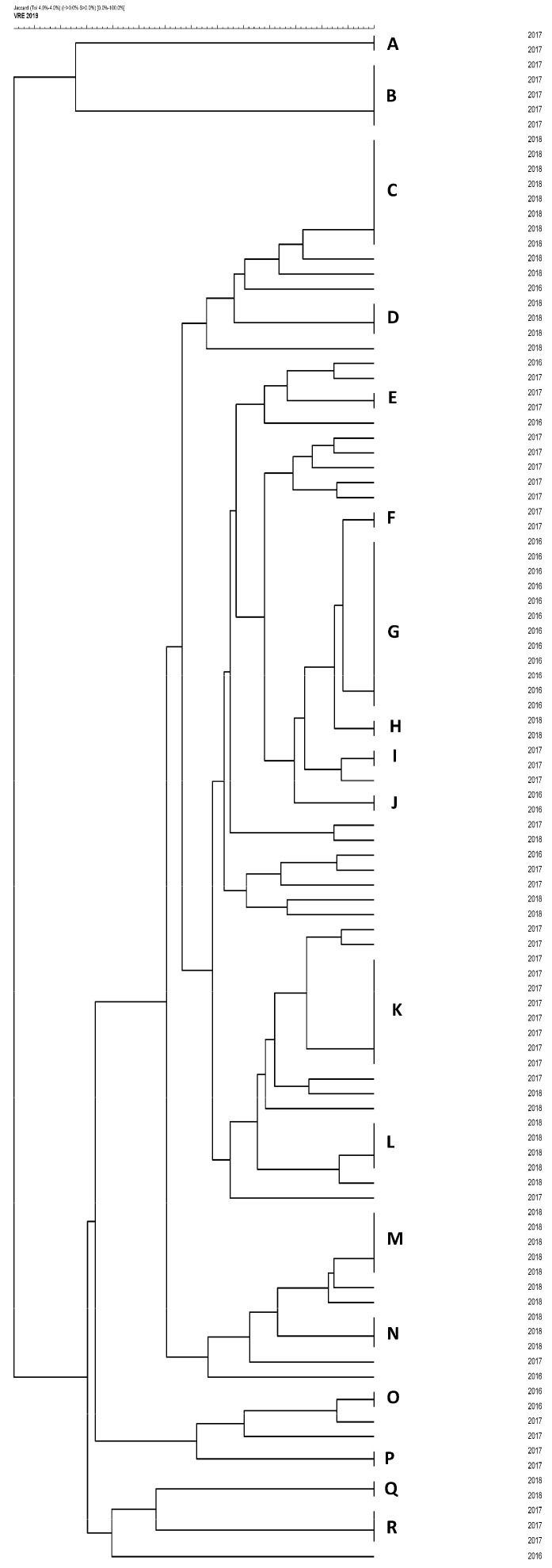
Pulsed-field gel electrophoresis (PFGE) dendrogram of 103 vancomycin-resistant *E. faecium* isolates. The letters represent 18 clonal groups of VRE (A–R) based on their similarity.

**Figure 2 antibiotics-09-00785-f002:**
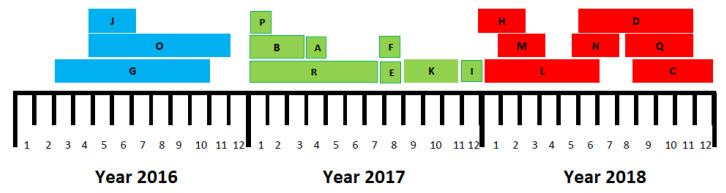
Timeline of the prevalence of VRE clones throughout the study duration. The letters represent 18 clonal groups of VRE (A–R) based on their similarity.

**Table 1 antibiotics-09-00785-t001:** Susceptibility of vancomycin-resistant enterococci (VRE) to selected antibiotics (percentages).

Year/Antibiotic	AMP	TIG	TET	TEI	FUR	LNZ
**2016**	0	100	71	0	100	100
**2017**	0	88	43	0	84	100
**2018**	0	100	35	0	48	100
**2016–2018**	0	96	48	0	76	100

Legend: AMP—ampicillin, TIG—tigecycline, TET—tetracycline, TEI—teicoplanin, FUR—nitrofurantoin, LNZ—linezolid.

**Table 2 antibiotics-09-00785-t002:** PCR primers and products for detecting *van* genes and virulence factor genes.

Gene	Sequence (5′→3′)	Size (bp)	Reference
***van* Genes**			
***vanA***	GGGAAAACGACAATTGC	732	[42]
	GTACAATGCGGCCGTTA		
***vanB***	ATGGGAAGCCGATAGTC	635	
	GATTTCGTTCCTCGACC		
***vanC-1***	GGTATCAAGGAAACCTC	822	
	CTTCCGCCATCATAGCT		
***vanC2-C3***	CTCCTACGATTCTCTTG	439	
	CGAGCAAGACCTTTAAG		
**Virulence Factor Genes**			
***asa1***	GCACGCTATTACGAACTATGA	375	[43]
	TAAGAAAGAACATCACCACGA		
***gelE***	TATGACAATGCTTTTTGGGAT	213	
	AGATGCACCCGAAATAATATA		
***cylA***	ACTCGGGGATTGATAGGC	688	
	GCTGCTAAAGCTGCGCTT		
***esp***	AGATTTCATCTTTGATTCTTGG	510	
	AATTGATTCTTTAGCATCTGG		
***hyl***	ACAGAAGAGCTGCAGGAAATG	276	
	GACTGACGTCCAAGTTTCCAA		

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
