# Peer review of "Analysis of Vancomycin-Resistant Enterococci in Hemato-Oncological Patients"

_antibiotics, 2020, doi:10.3390/antibiotics9110785_

Round 1

Reviewer 1 Report

Enterococci, and specifically vancomycin-resistant enterococci, pose a significant medical problem. These bacteria cause healthcare associated infections, most often in long-term hospital patients, those with underlying health issues or those receiving broad-spectrum antibiotics.

The current studying assessed the presence of innately present vancomycin-resistant enterococci (VRE) in the gut flora of hemato-oncology patients. A thorough investigation of the VRE species present in the patient cohort was conducted: phenotypic and genotypic analyses, antibiotic susceptibility, expression of resistance and virulence factor genes, the clonal relationships and interplay between different VRE strains identified and the timeline of emergence.

The authors should be commended on their comprehensive and well written manuscript, congratulations.

I only have one comment…

The authors report that 7.3% of hemato-oncological patients harbour VRE in their gastrointestinal tract. 7.3% seems low given the other studies that are cited in the text:

  • Increased from 3% before to 59% during hospital stay, Heisel et al
  • 26% and 31% colonization after antibiotic treatment, Paterson et al.
  • Meta-analysis of 45 studies – average of 20% colonization, Alevizakos et al
  • 74% colonization in AML patients, Ballo et al

Can the authors comment on why the percentage of patients carrying VRE in the present study is considerably lower compared with previous studies.

Author Response

Reply to review report

The authors thank the reviewer for the comments; the manuscript has been modified based on them.

Comment: The authors report that 7.3% of hemato-oncological patients harbour VRE in their gastrointestinal tract. 7.3% seems low given the other studies that are cited in the text:

  • Increased from 3% before to 59% during hospital stay, Heisel et al
  • 26% and 31% colonization after antibiotic treatment, Paterson et al.
  • Meta-analysis of 45 studies – average of 20% colonization, Alevizakos et al
  • 74% colonization in AML patients, Ballo et al

Can the authors comment on why the percentage of patients carrying VRE in the present study is considerably lower compared with previous studies.

Reply: In our study, low prevalence of VRE was demonstrated in the GIT of patients with hemato-oncological disease compared to other studies reporting prevalence rates of 3% to 59%. This may be explained by very low prevalence of VRE (below 1%, unpublished data) in the community population of the region from where the monitored patients come. Another important reason is implementation of bacteriological surveillance in hemato-oncological patients, including active search for bacterial strains with dangerous resistance phenotypes (e.g. VRE, ESBL-, AmpC- and KPC-positive enterobacteria) in the microflora of the upper respiratory tract and GIT. In case of positive detection, barrier measures are strictly observed in these patients.

Reviewer 2 Report

In this study, Hricova and co-authors investigate the properties of Vancomycin-resistant enterococci (VRE) in the gut of hematological patients with cancer. The authors analyzed the stool and body fluid samples of hospitalized patients and found that:

  1. 7% of patients had VRE presence in the gut.
  2. Extracellular surface protein was the most common isolated virulence factor in the analyzed samples.
  3. Different clones of VRE persisted for different duration during the hospitalization stay in the analyzed patient cohort, suggesting that no one strain was causative for VRE prevalence in the patient population.

The authors could shorten the 'Introduction' section significantly, since most of the text is descriptive in nature and is not necessary for the reader to understand the results presented in the manuscript. Similarly, the manuscript should be edited to a more scientific writing style compared to the newspaper style that it is currently presented in.

The authors should perform a better analysis of the presence of different virulence genes and their correlation with the different clones during the duration of the study. 

The authors should also assess whether any novel and previously undescribed genes are present in the VRE clones from the samples collected. 

Finally, only 7% of patients have the presence of VRE in the gut. What is the reason for the low prevalence, and if possible, how does the length of hospital stay correlate with the amount of VRE positivity in the current study?

Author Response

Reply to review report

The authors thank the reviewer for the comments; the manuscript has been modified based on them.

Comment: The authors could shorten the 'Introduction' section significantly, since most of the text is descriptive in nature and is not necessary for the reader to understand the results presented in the manuscript. 

Reply: The 'Introduction' section has been shortened and edited.

Comment: The authors should perform a better analysis of the presence of different virulence genes and their correlation with the different clones during the duration of the study.

Reply: The correlation between different virulence genes and different clones was added. No differences were observed between the individual clones in the presence of the observed virulence factors.

Comment: The authors should also assess whether any novel and previously undescribed genes are present in the VRE clones from the samples collected. 

Reply: No novel and previously undescribed genes were detected in the VRE clones.

Comment: Finally, only 7% of patients have the presence of VRE in the gut. What is the reason for the low prevalence, and if possible, how does the length of hospital stay correlate with the amount of VRE positivity in the current study?

Reply: In our study, low prevalence of VRE was demonstrated in the GIT of patients with hemato-oncological disease compared to other studies reporting prevalence rates of 3% to 59%. This may be explained by very low prevalence of VRE (below 1%, unpublished data) in the community population of the region from where the monitored patients come. Another important reason is implementation of bacteriological surveillance in hemato-oncological patients, including active search for bacterial strains with dangerous resistance phenotypes (e.g. VRE, ESBL-, AmpC- and KPC-positive enterobacteria) in the microflora of the upper respiratory tract and GIT. In case of positive detection, barrier measures are strictly observed in these patients. No correlation between the length of hospital stay and the amount of VRE positivity was confirmed.

Reviewer 3 Report

Detailed Comments for Authors:

The manuscript presents a logically built, well conducted multi-year study.

Below are the more detailed questions from this Reviewer:

  1. Page 6, Line 189: What were the specific antibiotics used in the microdilution method?

Q1: Are these infections considered nosocomial or was it possible that the patients were harboring the bacteria already at admission?

Q2: How close were the patients or their rooms to each other with long-term infections of the same VRE (G, R, or L)?

Q3: Was there any indication of airborne transmission, especially for the transition of the GIT VRE infection to pneumonia? Were surfaces in the rooms tested for the presence of VRE?

Q4: What happened when the isolates causing long-term infections (G, R, L) weren’t detected anymore? Did the patients leave or was there a change in the environmental parameters (remodeling of the rooms, new ventilation etc.)?

Another interesting result of the study could be the analysis of the accompanying or cohabitant isolates or microbiomes identified in the patients’ samples.

Q5: Were any of the other isolates detected as identical in other patients? What was the level of genetic difference between strains?

Q6: What would be the recommendation of the Authors for mitigation measures to reduce the number of VRE infections in the hospital rooms?

Author Response

Reply to review report

The authors thank the reviewer for the comments; the manuscript has been modified based on them.

Comment 1: Page 6, Line 189: What were the specific antibiotics used in the microdilution method?

Reply: The tested antibiotics were added to the text.

Q1: Are these infections considered nosocomial or was it possible that the patients were harboring the bacteria already at admission?

Reply: In our study, people with hemato-oncological disease were only carriers of VRE in the GIT; they had no bacterial infection at the time of VRE isolation.

Q2: How close were the patients or their rooms to each other with long-term infections of the same VRE (G, R, or L)?

Reply: Those patients stayed in the same ward of the Department of Hemato-Oncology. As there are two-bed rooms in this ward, it is clear that an identical vancomycin-resistant strain of E. faecium spread among patients in the same room and, at the same time, among patients from different rooms. The mode of transmission was not analyzed in our study, but it can be assumed that the spread was through the medical staff, as previously demonstrated in the Department of Hemato-Oncology in a 2006 study by Kolář et al. (citation 19 in our article).

Q3: Was there any indication of airborne transmission, especially for the transition of the GIT VRE infection to pneumonia? Were surfaces in the rooms tested for the presence of VRE?

Reply: Surfaces in the rooms were not tested. No VRE pneumonia was detected during the study. There is no indication of airborne transmission.

Q4: What happened when the isolates causing long-term infections (G, R, L) were not detected anymore? Did the patients leave or was there a change in the environmental parameters (remodeling of the rooms, new ventilation etc.)?

Reply: It may be assumed that the patients were gradually discharged from the Department of Hemato-Oncology. There were no changes in the environmental parameters such as remodeling of the rooms or new ventilation.

Comment: Another interesting result of the study could be the analysis of the accompanying or cohabitant isolates or microbiomes identified in the patients’ samples.

Reply: A similar study on enterobacteria with the production of broad-spectrum beta-lactamases has already been performed. This study identified patients diagnosed with urinary tract and bloodstream infections caused by ESBL-positive strains of Klebsiella pneumoniae contained in the GIT microflora.

(Kolář M., Htoutou Sedláková M., Pudová V., et al. Incidence of fecal Enterobacteriaceae producing broad-spectrum beta-lactamases in patients with hematological malignancies. Biomed Papers. 2015, 159:100-103.)

Q5: Were any of the other isolates detected as identical in other patients? What was the level of genetic difference between strains?

Reply: Only isolated VRE were analyzed during the study.

Q6: What would be the recommendation of the Authors for mitigation measures to reduce the number of VRE infections in the hospital rooms?

Reply: Active search for VRE using selective agar, hygienic-epidemiological measures and their observance by the medical staff, sanitation of rooms and toilets.

Round 2

Reviewer 3 Report

This Reviewer would like to thank the Authors for addressing the Review comments in detail and making relevant amendments in the manuscript.